# Anti-Atherosclerotic Activity of (3*R*)-5-Hydroxymellein from an Endophytic Fungus *Neofusicoccum parvum* JS-0968 Derived from *Vitex rotundifolia* through the Inhibition of Lipoproteins Oxidation and Foam Cell Formation

**DOI:** 10.3390/biom10050715

**Published:** 2020-05-05

**Authors:** Jae-Yong Kim, Soonok Kim, Sang Hee Shim

**Affiliations:** 1College of Pharmacy, Duksung Woman’s University, Seoul 01369, Korea; kjaey0331@naver.com; 2Biological Resources Assessment Division, National Institute of Biological Resources, Incheon 22689, Korea; sokim90@korea.kr

**Keywords:** (3*R*)-5-hydroxymellein, low-density lipoprotein, high-density lipoprotein, oxidation, foam cell, atherosclerosis

## Abstract

An endophytic fungus, *Neofusicoccum parvum* JS-0968, was isolated from a plant, *Vitex rotundifolia*. The chemical investigation of its cultures led to the isolation of a secondary metabolite, (3*R*)-5-hydroxymellein. It has been reported to have antifungal, antibacterial, and antioxidant activity, but there have been no previous reports on the effects of (3*R*)-5-hydroxymellein on atherosclerosis. The oxidation of lipoproteins and foam cell formation have been known to be significant in the development of atherosclerosis. Therefore, we investigated the inhibitory effects of (3*R*)-5-hydroxymellein on atherosclerosis through low-density lipoprotein (LDL) and high-density lipoprotein (HDL) oxidation and macrophage foam cell formation. LDL and HDL oxidation were determined by measuring the production of conjugated dienes and malondialdehyde, the amount of hyperchromicity and carbonyl content, conformational changes, and anti-LDL oxidation. In addition, the inhibition of foam cell formation was measured by Oil red O staining. As a result, (3*R*)-5-hydroxymellein suppressed the oxidation of LDL and HDL through the inhibition of lipid peroxidation, the decrease of negative charges, the reduction of hyperchromicity and carbonyl contents, and the prevention of apolipoprotein A-I (ApoA-I) aggregation and apoB-100 fragmentation. Furthermore, (3*R*)-5-hydroxymellein significantly reduced foam cell formation induced by oxidized LDL (oxLDL). Taken together, our data show that (3*R*)-5-hydroxymellein could be a potential preventive agent for atherosclerosis via obvious anti-LDL and HDL oxidation and the inhibition of foam cell formation.

## 1. Introduction

Atherosclerosis, a widespread and chronic progressive arterial disease characterized by lipid accumulation and chronic inflammation in the arterial wall, is the main cause of coronary heart disease, cerebrovascular disease, cardiac ischemia, ischemic stroke, and peripheral vascular disease, and it is a leading major cause of death in the industrial world [1,2]. Atherosclerosis is caused by many risk factors such as dyslipidemia, hypertension, diabetes, cigarette smoking, alcohol, family history, and lipoproteins modification [3]. Among them, lipoproteins modification is well established to promote an impressive role in the initiation and development of atherosclerosis [4].

Two major lipoproteins in human plasma, low-density lipoprotein (LDL) and high-density lipoprotein (HDL), are known as the main carriers for delivering cholesterol transportation and maintaining cholesterol homeostasis in the body [5]. However, they can be easily modified by oxidation, glycosylation, lipolysis, smoking, and aging [6,7]. Especially, oxidized LDL (oxLDL) acts as the highest risk factor for the progression of atherosclerosis by up-regulating the adhesion of monocytes to endothelial cells, a proliferation of vascular smooth muscle cells (VSMCs), and the induction of foam cell formation of macrophages [8]. In RAW264.7 mouse macrophages, oxLDL induced the increase of several pro-inflammatory cytokines including such as tumor necrosis factor (TNF)-α and interleukin (IL)-6, and the up-regulation of scavenger receptor including lectin-like oxidized low-density lipoprotein receptor-1 (LOX-1) and the cluster of differentiation 36 (CD36) [9,10]. In addition, cells involved in atherosclerosis such as lymphocytes, and monocyte-derived macrophages, endothelial cells, and smooth muscle cells become cytotoxic by oxLDL [11,12]. Furthermore, plasma oxLDL levels in patients with coronary heart disease (CHD), acute myocardial infarction (AMI), metabolic syndrome, diabetes mellitus (DM), nonalcoholic steatohepatitis (NASH), and hypertension were higher than in the control group [13,14,15].

High-density lipoprotein (HDL) is generally well known as one of the most important anti-atherogenic factors. Numerous cohort studies have demonstrated the close relationship between a low serum HDL level and a high risk of coronary heart disease (CHD), and it also revealed that a high level of HDL-C was strongly related to the inhibition of CHD in an independent way [16,17]. Besides, numerous previous studies have demonstrated that HDL plays an important role in decreasing the risk of atherosclerosis through anti-oxidative, reverse cholesterol transport, anti-inflammatory, antithrombotic properties, anti-LDL oxidation, and endothelial cell maintenance functions [18,19,20,21]. However, oxidized HDL causes the risk of atherosclerosis through the loss of their anti-atherogenic functions and increased cytotoxicity, reactive oxygen species (ROS) production, and lipid accumulation [22,23]. Furthermore, it has been reported that plasma oxHDL levels in patients with DM and CHD are higher than in healthy humans [24,25]. Therefore, inhibitors of LDL and HDL oxidation could be a good strategy to prevent heart diseases.

Endophytes are microbes that spend the whole or parts of their life cycle within a plant and have a symbiotic relationship without causing apparent pathogenic symptoms [26]. According to many reports, endophytes are potential sources for secondary metabolites with a variety of biological activities including antioxidant, antifungal, antiviral, and anticancer [27]. In our continuous studies on the inhibitory activities of natural compounds against lipoproteins’ modification, an endophytic fungus was isolated from a plant *Vitex rotundifolia*, and its chemical investigation was conducted in this study. The strain was cultivated on potato dextrose agar (PDA) media for three weeks and extracted with organic solvents. A series of chromatographies were conducted for the isolation of a secondary metabolite from the fungal extracts. The isolated metabolites were identified to be (3*R*)-5-hydroxymellein by interpretation of its spectral data. It was first reported in 1990, and only a few studies have revealed its biological activities such as antifungal, antibacterial, antioxidant, the protection of UVB, and inhibition of melanogenesis [28,29,30]. However, there are no studies on the anti-atherosclerotic function of (3*R*)-5-hydroxymellein through lipoproteins (LDL and HDL) oxidation and the foam cell formation of macrophages. Herein, we first investigate the inhibitory effects of (3*R*)-5-hydroxymellein against lipoproteins’ oxidation and foam cell formation.

## 2. Materials and Methods

### 2.1. Reagents

Tetramethylsilane was obtained from Cambridge Isotope Laboratories (Tewksbury, MA, USA). copper sulfate (CuSO_4_), 1,1,3,3-tetramethoxypropane, dialysis tubing cellulose membrane, bovine serum albumin (BSA), folin phenol reagent, dichlorofluorescein diacetate (DCFH-DA), and dimethyl sulfoxide were purchased from Sigma Aldrich (St. Louis, MO, USA). Coomassie brilliant blue R-250 and trichloroacetic acid (TCA) were purchased from Tokyo Chemical Industry (Tokyo, Japan). Sodium chloride, sodium bromide, and acetic acid were obtained Merck (Darmsstadt, Germany).

### 2.2. General Experimental Procedures

Low-resolution electrospray ionization mass spectrometry (LR-ESI−MS) results were collected using a Waters micromass ZQ mass spectrometer (Waters, Milford, MA, USA). Nuclear magnetic resonance (NMR) was conducted using a Varian NMR spectrometer (^1^H, 500 MHz; ^13^C, 125 MHz; Varian, CA, USA) with tetramethylsilane, and chemical shifts were displayed as δ values. High-performance liquid chromatography (HPLC) was subjected to a Waters system equipped with a 600 controller, UV detector (996 photodiode array), and Luna 5 μm C18(2) 100 Å column (250 mm × 10 cm, Phenomenex, CA, USA). Open column chromatography was performed with silica gel (Merck, Germany). For thin-layer chromatography (TLC) analysis, 60 F254 and RP-18 F254S plates (Merck, Germany) were used, and visualization of the plates was performed using UV light and 10% sulfuric acid solution.

### 2.3. Isolation and Cultivation of Fungus Strain

The fungal strain (JS-0968) was isolated from *Vitex rotundifolia*, which was collected from a seashore of Jeju island, South Korea in September 2012. Stem tissue was chopped into small pieces and sterilized using 2% sodium hypochlorite and 70% ethanol, followed by washing with sterilized distilled water. Fungal strains were cultured on malt extract agar (MEA, Difco) supplemented with 50 ppm of kanamycin, chloramphenicol, and Rose Bengal at 22 °C for 7 days. The growing colony was transferred to fresh potato dextrose agar (PDA medium (24 g of PD broth with 18 g of agar to 1 L of sterilized distilled water). The fungal strain was identified based on the BLAST (Basic Local Alignment Search Tool) best hit of ITS (Internal Transcribed Spacers) sequences, and it was stored on glycerol stocks (20% *w/v* in H_2_*O*) in a liquid nitrogen tank at the Wildlife Genetic Resources Bank at the National Institute of Biological Resources (Incheon, Korea). The JS-0968 strain was cultured on a PDA plate (100 mm Petri dish) for 7 days at room temperature, and they were cut into small pieces (10 × 10 mm). Subsequently, these pieces were inoculated into 500 mL Erlenmeyer flasks containing 67% of solid rice medium (80 g rice/120 mL sterilized distilled water) on solid rice medium and incubated at room temperature for 30 days.

### 2.4. Extraction and Isolation

The cultures were extracted with ethyl acetate to afford 15.7 g of the extract. The crude extract was separated using vacuum liquid column (VLC) chromatography on silica gel into 13 fractions (fractions A−M) with elution of *n*-hexane/acetone/MeOH (100:0:0, 70:1:0, 50:1:0, 40:1:0, 30:1:0, 20:1:0, 10:1:0, 7:1:0, 5:1:0, 3:1:0, 1:1:0, 1:1:0.2, 0:0:100; 3 L each). Fraction I was further subjected to reversed-phase column chromatography (H_2_O/MeOH, 60:40 → 0:100, *v*/*v*; 300 mL each) to get nine fractions (fraction I1-I9). Fractions I2 was further subjected to semi-preparative HPLC on reverse phase to afford 3-(*R*)-hydroxymellein (3 mg) eluted with H_2_O/acetonitrile (50:50 → 35:65, *v*/*v*) gradient solvents.

(3*R*)-5-hydroxymellein ^1^H NMR (DMSO-*d*_6_, 500 MHz) *δ*_H_ 10.39 (1H, s, 5-OH), 9.44 (1H, s, 8-OH), 7.07 (1H, d, *J* = 9.0 Hz, H-7), 6.72 (1H, d, *J* = 9.0 Hz, H-6), 4.73 (1H, m, H-3), 3.07 (1H, dd, *J* = 17.0, 3.5 Hz, H-4a), 2.59 (1H, dd, *J* = 17.0, 11.5 Hz, H-4b), 1.43 (3H, d, *J* = 6.5 Hz, 3-CH_3_); (+)ESIMS at *m/z* 195.0 [M + H]^+^.

### 2.5. Isolation of LDL and HDL from Human Plasma

This study was approved by the Research Ethics Review Committee of Duksung Women’s University (IRB No. 2020-004-006-B). Human whole blood was obtained from young and healthy male volunteers. The blood was collected at Vacutainer plastic tubes (BD sciences, Franklin Lakes, NJ, USA), and plasma was separated by high-speed centrifugation for 10 min at 4 °C at 1690× *g* (5810R; Eppendorf, Hamburg, Germany). LDL (1.019–1.063 g/mL) and HDL (1.125–1.225 g/mL) were separated from the plasma by sequential density ultracentrifugation [31]. Briefly, ultracentrifugation was run at 235,000× *g* for 22 hr at 10 °C using an LE-80 (Beckman, CA, USA). Isolated LDL and HDL were dialyzed overnight against Tris buffer (10 mM Tris–HCl, 140 mM NaCl, 5 mM ethylenediaminetetraacetic acid (EDTA), pH 7.4) at 4 °C for 24 h. The protein concentration of LDL and HDL was quantified according to the Lowry method with slight modification [32].

### 2.6. Conjugated Dienes (CD) Formation

For measuring the dienes production by the chain reaction of lipid peroxidation, LDL (50 μg protein/mL) and HDL (200 μg protein/mL) were incubated with CuSO_4_ (final concentration, 10 μM), respectively, under the presence or absence of (3*R*)-5-hydroxymellein (final concentrations, 10, 20, and 40 μM, respectively) and quercetin (final concentration, 40 μM) as the control in a medium containing phosphate buffer (10 mM, pH 7.4). During incubation, kinetics for CD formation were determined by measuring the absorbance at 234 nm with every 2 min interval for a total duration of 180 min using a UV-VIS Spectra Max 190 spectrophotometer (Molecular Devices, CA, USA) [33].

### 2.7. Thiobarbituric Acid Reactive Substances (TBARS) Formation

Native LDL and HDL (500 μg protein/mL, respectively) with 10 μM CuSO_4_ in the presence or absence of (3*R*)-5-hydroxymellein (final concentrations, 10, 20, and 40 μM, respectively) and quercetin (final concentration, 40 μM) were incubated for 4 h at 37 °C, and 0.5 mM EDTA was added to terminate oxidation. Afterwards, 20% trichloroacetic acid (TCA) was added to samples, and then the mixtures were incubated with 0.67% thiobarbituric acid (TBA). The final mixture was placed in a water bath for 20 min at 90 °C, and they were cooled at 4 °C and centrifuged for 20 min at 848× *g*. The optical density of the supernatant was measured at 532 nm with UV-VIS Spectra Max 190 spectrophotometer (Molecular Devices, CA, USA) to determine the amount of TBARS.

### 2.8. Change of Relative Electrophoretic Mobility (REM)

The electrophoretic mobilities of native or oxidized LDL and HDL were performed by agarose gel electrophoresis [34]. First, 10 µL each of LDL and HDL (4 µg of protein) were loaded onto a 0.5% agarose gel and electrophoresed in an electrophoretic system (100 V for 40 min in TAE buffer; 40 mM Tris-Acetate, 1 mM EDTA, pH 8.0). After electrophoresis, the gels were fixed using a solution (ethanol/acetic acid/distilled water, 60:10:30, *v/v/v*), and they were dried for 1 h at 80 °C, followed by staining with coomassie brilliant blue R250 (0.15%) [34].

### 2.9. Determination of Apolipoproteins Modification by SDS-PAGE

After oxidation, Laemmli sample buffer and 2-mercaptoethanol (15:1, *v/v*) was applied to each LDL and HDL sample for denaturation for 5 min at 90 °C. SDS polyacrylamide gel electrophoresis (6% and 12% SDS-PAGE, respectively) was conducted to detect the apoB-100 fragmentation and the apoA-I aggregation. Afterwards, the gels were stained with Coomassie blue (0.15%) to visualize apoB-100 and apoA-I.

### 2.10. Measurement of UV Absorbance

After oxidation, the UV absorbance of natural or oxidized LDL and HDL was measured at 280 nm using a Spectra Max 190 UV-visible spectrophotometer (Molecular Devices, CA, USA) to evaluate the change of hyperchromicity, which was calculated by the following equation: percentage of Hyperchromicity at 280 nm = [Absorbance (Abs) of oxidized sample – Abs of native or compound treated sample)/Abs of oxidized sample] × 100 [35].

### 2.11. Measurement of Protein-Bound Carbonyl Groups

Protein-bound carbonyls were measured using the carbonyl-specific reagent 2,4-dinitrophenylhydrazine (DNPH) [36,37]. First, 250 μL of native or oxidized samples were mixed with 350 μL of DNPH solution (7 mM DNPH in 2 N HCl). After 1 h at room temperature for the formation of DNP hydrazones, 20% trichloroacetic acid was added. The mixture was centrifuged with 9425× *g* for 10 min, and then the pellet was washed three times with 1 mL of ethanol/ethyl acetate (1:1. *v*/*v*) to remove unreacted DNPH. Afterwards, the pellet was dissolved by 600 μL of a 6 M guanidine hydrochloride (GdnHCl) in 20 mM phosphate buffer. The DNPH samples were read at 379 nm using a UV-visible Spectra Max 190 spectrophotometer (Molecular Devices, CA, USA) and the carbonyl concentration of each sample was calculated using a ε379 nm = 22,000 M^−1^∙cm^−1^ [38].

### 2.12. Measurement of Dichlorofluorescein (DCF) Fluorescence

HDL (500 μg/mL) was pretreated with 40 μM of (3*R*)-5-hydroxymellein or quercetin for 4 h at 37 °C. Subsequently, the samples were dialyzed during 24 h at 4 °C against phosphate buffer (10 mM, pH 7.4). LDL (170 μg/mL) oxidation was induced by 0.5 μM of CuSO_4_ for 4 h at 37 °C under the absence or presence of HDL (60 μg/mL). Subsequently, copper ions were removed through dialysis. Then, 2 mg of dichlorofluorescein diacetate (DCFH-DA) was dissolved in 1 mL of methanol, and then the mixture was incubated in the dark state (at room temperature, 30 min). Protein mixture (50 μL) was mixed with DCFH (10 μL) and normal saline (460 μL). Final samples were incubated for 2 h at 37 °C in the dark. DCF fluorescence was measured by the fluorescence intensity (Ex = 485 nm, Em = 538 nm) using a Synergy 2 multi-mode microplate reader (BioTek, VT, USA) [39,40].

### 2.13. RAW 264.7 Cell Culture and Treatment

Mouse monocyte line, RAW 264.7 cells, were purchased from the Korean Cell Line Bank (Seoul, South Korea). The cells were maintained in Dulbecco’s modified Eagle’s medium (Hyclone, Utah, USA) supplemented with 10% fetal bovine serum (FBS) (Welgene, Gyeongsan, South Korea) and 1% penicillin/streptomycin (Invitrogen Co., NY, USA) at 37 ◦C in a humidified atmosphere of 5% CO_2_ incubator. To evaluate the anti-atherogenic effect, RAW264.7 cells were plated onto 24-well plates at a density of 2 × 10^5^ cells/well and allowed to attach for 24 h, and then the cells were pretreated with 3-(*R*)-hydroxymellin or quercetin (final concentration, 40 µM) for 4 h, followed by treatment with oxLDL (final 50 µg/mL) for 24 h.

### 2.14. Cell Viability Assay

The cytotoxicity of (3*R*)-5-hydroxymellein on RAW 264.7 cells was evaluated using an ELUS cell viability assay kit (Biosesang, SeongNam, Korea) according to the manufacturer’s instructions. Briefly, 1×10^4^ RAW 264.7 cells/well were cultured at 96-well plates for 24 h at 37 °C in a humidified atmosphere of 5% CO_2_ incubator. Subsequently, cells were treated with (3*R*)-5-hydroxymellein or quercetin (final concentration, 40 µM) for 24 h. Afterwards, 10 μL of 2-(2-Methoxy-4-nitrophenyl)-3-(4-nitrophenyl)-5-(2,4-disulfophenyl)-2H-tetrazolium Sodium Salt (WST-8) dye was add to each well, cells were incubated at 37 °C for 4 h, and the absorbance was determined at 450 nm using a Synergy 2 multi-mode microplate reader (BioTek, VT, USA).

### 2.15. Measurement of Foam Cell Formation

For Oil red O staining, the cells were washed three times with phosphate-buffered saline (PBS) and fixed with 10% formaldehyde (*v*/*v*) for 20 min at room temperature. After fixation, the cells were rinsed three times with PBS, and then the cells were stained with a 60% Oil Red O solution (Sigma-Aldrich, MO, USA) for 30 min at 37 °C. Subsequently, the cells were washed with ddH_2_O to remove the background dye and the foam cell formation observed and photographed under an inverted light microscope (TE200; Nikon, Tokyo, Japan), followed by computer image analysis using Image J software.

### 2.16. Statistical Analyses

Data were presented as the mean ± standard deviation (SD). Statistical significance was determined using analysis of variance (ANOVA), followed by Bonferroni multiple testing correction. A *P*-value of less than 0.05 was considered statistically significant.

## 3. Results

### 3.1. Structure Elucidation of (3R)-5-Hydroxymellein

First, (3*R*)-5-hydroxymellein was isolated from cultures of an endophytic fungus *Neofusicoccum parvum* JS-0968, and it was identified by comparing its spectra data with reported literature values [28].

### 3.2. Effects of (3R)-5-Hydroxymellein on Conjugated Dienes and TBARS Formation from LDL and HDL Oxidation

Lipid peroxidation in LDL and HDL was evaluated by measuring the total amounts of CD and TBARS production. Oxidized LDL and HDL mediated by copper ion increased CD formation and accelerated lag phase compared to the native state (Figure 1A and Figure 2A). However, LDL and HDL treated with (3*R*)-5-hydroxymellein (at 10, 20, and 40 µM) significantly decreased CD formation and delayed lag time in copper ion-induced LDL and HDL oxidation (Figure 1A and Figure 2A). In addition, the MDA level of oxLDL and oxHDL was found to be significantly higher than that of native LDL and HDL (Figure 1B and Figure 2B). On the contrary, (3*R*)-5-hydroxymellein significantly inhibited MDA formation by LDL and HDL oxidation. LDL treated with (3*R*)-5-hydroxymellein (at 20 and 40 µM) reduced MDA formation up to 16% and 92% compared to oxLDL, respectively (Figure 1B). Similarly, HDL treated with (3*R*)-5-hydroxymellein (at 10, 20, and 40 µM) inhibited the formation of MDA up to 47%, 92%, and 94% compared to oxHDL, respectively (Figure 2B). Quercetin was used as a positive control, which results were similar to those treated with 40 µM of 3-(*R*)-hydroxymellein.

### 3.3. Effects of (3R)-5-Hydroxymellein on UV Absorption and Carbonyl Content of Oxidized LDL and HDL

The increases of hyperchromicity and carbonyl contents were known as the markers of LDL and HDL oxidation. In Figure 1C,D and Figure 2C,D, oxLDL and oxHDL were shown to be remarkably higher than that of native LDL and HDL. However, LDL treated with (3*R*)-5-hydroxymellein (at 40 µM) significantly reduced hyperchromicity and carbonyl contents up to 79% and 85% compared to oxLDL, respectively (Figure 1C,D). In addition, in HDL treated with (3*R*)-5-hydroxymellein (at 40 µM), the increase of hyperchromicity and carbonyl content was significantly suppressed (96% and 74%, respectively) compared to oxidized HDL (Figure 2C,D). Surprisingly, even at a low concentration of (3*R*)-5-hydroxymellein (10 and 20 µM), it powerfully inhibited the elevation of hyperchromicity (86% and 89%, respectively) and carbonyl content (50% and 73%, respectively) (Figure 2C,D).

### 3.4. Effects of (3R)-5-Hydroxymellein on Change of Charge and apoB-100 Degradation on LDL Oxidation

The change of electrical charge on LDL was evaluated by agarose gel electrophoresis analysis. The relative electrophoretic mobility (REM) of oxidized LDL increased almost two-fold compared to the native LDL (Figure 3A,B). However, treatment of (3*R*)-5-hydroxymellein (at 10, 20, and 40 µM) to native LDL significantly decreased in a concentration-dependent manner compared to the oxidized LDL (Figure 3A,B). The inhibitory effects of (3*R*)-5-hydroxymellein (at 40 μM) on the apolipoprotein B-100 (apoB-100) degradation of LDL was analyzed using SDS-PAGE (Figure 3C). The apoB-100 band in oxidized LDL disappeared because of the protein degradation caused by copper ion. Despite copper ion-induced oxidation, LDL treated with (3*R*)-5-hydroxymellein (at 40 μM) remarkably recovered apoB-100 degradation and was similar to the natural state (Figure 3C).

### 3.5. Effects of (3R)-5-Hydroxymellein on apoA-I Aggregation and anti-LDL Oxidation in oxHDL

The multimeric pattern of apoA-I shown in oxidized HDL indicated that copper ions accelerated the aggregation of apoA-I (Figure 4A). However, HDL treated with (3*R*)-5-hydroxymellein (at 10, 20, and 40 μM) significantly reduced apoA-I aggregation, and surprisingly their apoA-I pattern was similar to the native HDL (Figure 4A). The inhibitory effects of HDL on LDL oxidation was performed by measuring DCF fluorescence. Oxidized LDL markedly elevated the DCF fluorescence compared to native LDL (Figure 4B). However, it was significantly reduced in oxLDL treated with native HDL (Figure 4B), which means that HDL inhibits the oxidation of LDL. Interestingly, HDLs treated with (3*R*)-5-hydroxymellein (at 40 μM) remarkably reduced DCF fluorescence more than the control (HDL alone) (Figure 4B).

### 3.6. Effect of (3R)-5-Hydroxymellein on Foam Cell Formation

Cell viability analysis showed that 40 µM (3*R*)-5-hydroxymellein had no toxicity in RAW264.7 cells (Figure 5B). Oil Red O staining was used to measure the effects of (3*R*)-5-hydroxymellein on lipid accumulation. In RAW264.7 cells treated with 50 μg/mL of oxidized LDL, the Oil red O staining area was significantly increased compared to native LDL, indicating that oxLDL effectively induces foam cell formation (Figure 5A,C). However, the addition of (3*R*)-5-hydroxymellein (at 40 µM) to oxLDL-induced foam cells remarkably inhibited the lipid accumulation (Figure 5A,C). Quercetin was used as a positive control, and the results were similar to those treated with 40 µM of (3*R*)-5-hydroxymellein.

## 4. Discussion

Over the past decade, many researches showed that secondary metabolites produced by endophytic fungi have diverse bioactivities including antiviral, antifungal, anticancer, and antibacterial properties. In this study, we isolated (3*R*)-5-hydroxymellein from an endophytic fungus *Neofusicoccum parvum* JS-0968 derived from *Vitex rotundifolia*. To date, few studies on the biological activity of (3*R*)-5-hydroxymellein have been reported. In particular, there has been no studies on its effects on lipoprotein oxidation and foam cell formation. Herein, we report the isolation of (3*R*)-5-hydroxymellein from an endophytic fungus and its inhibitory effects against the oxidation of human plasma LDL and HDL and formation of foam cells induced by oxLDL.

The measurement of conjugated dienes (CDs) and malondialdehyde (MDA) is the most widely used method to measure the lipid peroxidation of lipoproteins in vitro [41]. CDs, which are produced by rearrangement of double bonds in polyunsaturated fatty acids (PUFAs) of LDL/HDL, are generated in the early stage of the peroxidation of PUFA. The formation of CDs and its lag time are well-known indicators of lipoprotein oxidation and significantly associated with the risk of coronary heart disease [42]. MDA is one of the most established final products of lipid peroxidation, and its level is regarded as a biomarker for oxidative stress. Previous studies have reported that elevated serum MDA levels are associated with the initiation and development of atherosclerosis [43]. In our results, (3*R*)-5-hydroxymellein significantly decreased both CD generation and MDA formation on oxidized LDL and HDL mediated by copper ions (Figure 1A,B and Figure 2A,B). Even a low concentration of (3*R*)-5-hydroxymellein (at 10 μM) significantly inhibited the formation of MDA on oxidized HDL (Figure 2A,B). These results indicated that (3*R*)-5-hydroxymellein could inhibit lipid peroxidation via the inhibition of CD and MDA formation on copper-mediated oxidized LDL and HDL, and it is particularly more effective for HDL rather than LDL.

The increase in hyperchromicity reflects exposure to chromophoric aromatic residues via degradation and unfolding in oxidized proteins [44]. It has been reported that the hyperchromicity of oxidized LDL and hemoglobin increased significantly compared to the natural state [45,46]. In the present results, LDL treated with (3*R*)-5-hydroxymellein (at 40 µM) and HDL treated with (3*R*)-5-hydroxymellein (at 10, 20, and 40 µM) significantly reduced hyperchromicity (Figure 1C and Figure 2C). These findings indicated that (3*R*)-5-hydroxymellein prevented modification including the unfolding and fragmentation of oxidized LDL and HDL by copper ions.

The oxidation of proteins was known to cause the production of carbonyl groups, which could be used as a marker of oxidative stress damage [47]. Carbonyl contents were shown to increase in oxidized LDL and plasma of patients with type 2 diabetes in the previous reports [48,49]. Our present results showed that the level of carbonyl content of oxidized LDL and HDL was remarkably higher than that of the native state (Figure 1D and Figure 2D). However, the addition of (3*R*)-5-hydroxymellein (at 10, 20, and 40 μM) to LDL and HDL significantly reduced the carbonyl contents by oxidation. Our results demonstrated that (3*R*)-5-hydroxymellein is a good scavenger of carbonyl produced in oxidized LDL and HDL.

It is well known that the oxidation of LDL causes the neutralization of the positive charge of the ε-amino group of lysine in apoB-100, thereby increasing the migration to the negative charge [50]. Therefore, oxidized LDL has a higher negative charge than native LDL, so relative electrophoretic mobility (REM) is known to increase in proportion to the oxidation of LDL. In our results, the treatment of (3*R*)-5-hydroxymellein (at 10, 20, and 40 μM) remarkably recovered the increased REM of oxidized LDL (Figure 3A,B). In addition, apoB-100 fragmentation that modifies the conformation of LDL is one of the well-known features that occur during the oxidation process of LDL [51]. The fragmentation of apoB-100 can lead to increased recognition by a scavenger receptor as well as the reduced binding ability to the LDL receptor [52]. The treatment of 40 μM of (3*R*)-5-hydroxymellein recovered oxidative fragmentation of apoB-100 in a similar way to the native LDL (Figure 3C), suggesting its strong inhibitory activity on conformational change by LDL oxidation.

The main constituent of HDL, apolipoprotein A-I (ApoA-I), is the most critical role in reverse cholesterol transport via the macrophage ATP-binding cassette transporter A1 (ABCA1) and plays an important role in the various beneficial anti-atherogenic functions of HDL. Apo A-1 injured by oxidative procedures is strongly related to the induction of dysfunctional HDL with pro-atherosclerotic and pro-inflammatory properties [53]. The cross-linking of apoA-I has been reported to be a key feature of HDL oxidation [54,55]. The present results showed that oxidized HDL increased the formation of multimeric apoA-I (Figure 4A). However, apoA-I aggregation in oxidized HDL was significantly reduced HDL by the treatment of (3*R*)-5-hydroxymellein (Figure 4A). In particular, HDL treated with (3*R*)-5-hydroxymellein, even at low concentrations, showed an apoA-1 band similar to that of native HDL (Figure 4A). These results suggest that (3*R*)-5-hydroxymellein can reduce the formation of dysfunctional HDL through the significant inhibition of aggregation of apoA-I. Furthermore, a cell-free assay was also conducted to evaluate anti-LDL oxidation of HDL [39,40], which is the most important role of the anti-atherogenic function of HDL. The native HDL significantly reduced the fluorescent signal, suggesting that the native HDL strongly inhibited the LDL oxidation (Figure 4B). Interestingly, HDL treated with 40 μM of (3*R*)-5-hydroxymellein significantly decreased the fluorescent signal more than the control (HDL alone) (Figure 4B). These results indicated that (3*R*)-5-hydroxymellein contributes to the improvement of the anti-LDL oxidation function of HDL.

OxLDL is known to stimulate macrophages to transform into foam cells [56]. Thereby, the transformation of macrophage into foam cells is the important hallmark of the early stage in atherosclerotic plaque development. To measure foam cell formation from oxLDL, we used the Oil red O staining for lipid-laden macrophages. Lipid droplet formation by oxLDL has been reported to increase in RAW 264.7 macrophages. In our current results, the treatment of oxLDL significantly increased lipid droplets in RAW 264.7 cells (Figure 5A,C), which is in good agreement with the previously published data [56]. On the contrary, the co-treatment of (3*R*)-5-hydroxymellein and oxLDL markedly reduced the lipid accumulation up to 53%, compared with the control (oxLDL alone) group (Figure 5A,C). Besides, quercetin, a positive control, also showed a similar reduction of lipid accumulation to that of (3*R*)-5-hydroxymellein. The inhibitory effect of quercetin on foam cell formation induced by oxLDL has been reported [57]. Therefore, these results indicate the inhibitory effects of 3-(*R*)-hydroxymellein on foam cell formation via lipid accumulation.

## 5. Conclusions

To the best of our knowledge, this is the first study on the anti-atherosclerotic effects of (3*R*)-5-hydroxymellein. Our findings suggested that (3*R*)-5-hydroxymellein could be a good supplement to reduce the risk of atherosclerosis via inhibition of LDL and HDL oxidation and macrophage foam cell formation. Further studies will be required to deal with other possible anti-atherosclerosis-related mechanisms and in vivo efficacies for (3*R*)-5-hydroxymellein.

## Figures and Tables

**Figure 1 biomolecules-10-00715-f001:**
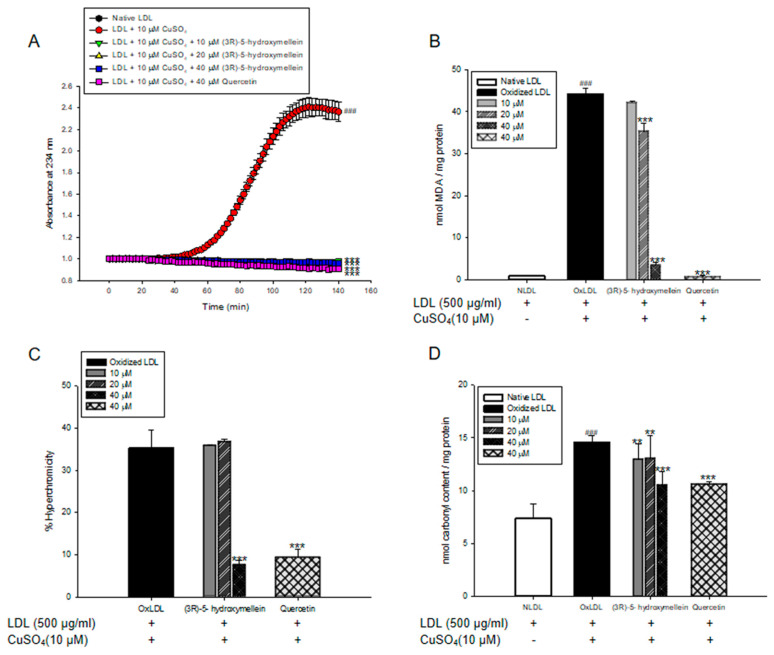
Effect of (3*R*)-5-hydroxymellein on Cu2+-induced low-density lipoprotein (LDL) oxidation. (**A**) Continuous monitoring of the conjugated diene levels by absorbance at 234 nm (A234) during 10 μM copper-ion-mediated LDL oxidation in the presence of (3*R*)-5-hydroxymellein. (**B**) Effects of (3*R*)-5-hydroxymellein (at 10, 20, and 40 μM) in Thiobarbituric Acid Reactive Substances (TBARS) production during LDL oxidation induced by CuSO4 for 4 h at 37 °C. (**C**) Percent of hyperchromicity (absorbance at 280 nm) of oxidized LDL (oxLDL) and LDL treated with (3*R*)-5-hydroxymellein (at 10, 20, and 40 μM). (**D**) Carbonyl content determination of native LDL, oxLDL, and LDL treated with (3*R*)-5-hydroxymellein (at 10, 20, and 40 μM). Quercetin used as a positive control. These data are expressed as the mean ± SD of three independent experiments. ### *p* < 0.001 vs. NLDL (native LDL); ** *p* < 0.01 vs. oxLDL (oxidized LDL); *** *p* < 0.001 vs. oxLDL (oxidized LDL).

**Figure 2 biomolecules-10-00715-f002:**
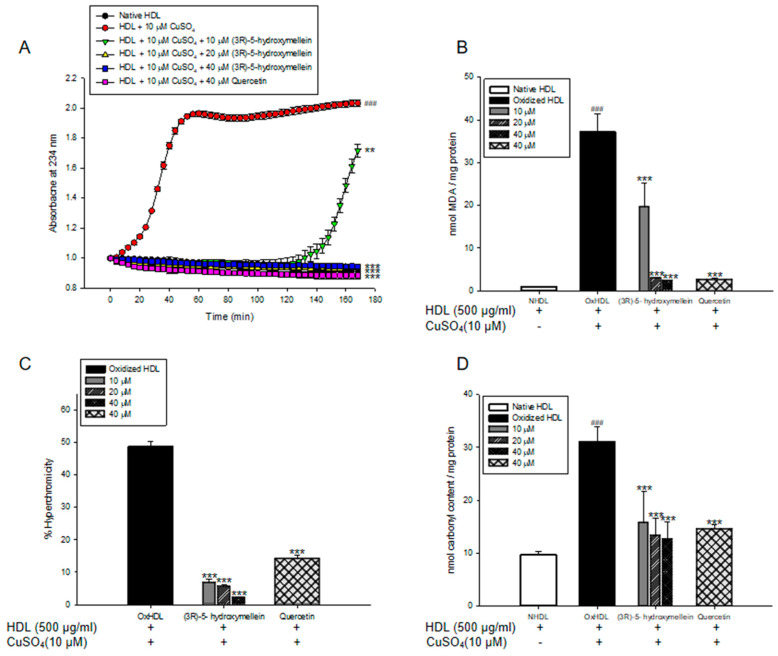
Effect of (3*R*)-5-hydroxymellein on Cu^2+^-induced high-density lipoprotein (HDL) oxidation. (**A**) Continuous monitoring of the conjugated diene levels by absorbance at 234 nm (A_234_) during 10 μM of copper-ion-mediated HDL oxidation in the presence of (3*R*)-5-hydroxymellein. (**B**) Effects of (3*R*)-5-hydroxymellein (at 10, 20, and 40 μM) in TBARS production during HDL oxidation induced by CuSO_4_ for 4 h at 37 °C. **(C)** Percent of hyperchromicity (absorbance at 280 nm) of oxHDL, and HDL treated with (3*R*)-5-hydroxymellein (at 10, 20, and 40 μM). (D) Carbonyl content determination of native HDL, oxHDL, and HDL treated with (3*R*)-5-hydroxymellein (at 10, 20, and 40 μM). Quercetin used as a positive control. These data are expressed as mean ± SD of three independent experiments. ^###^
*p* < 0.001 vs. NHDL (native HDL); *** *p* < 0.001 vs. oxHDL (oxidized LDL).

**Figure 3 biomolecules-10-00715-f003:**
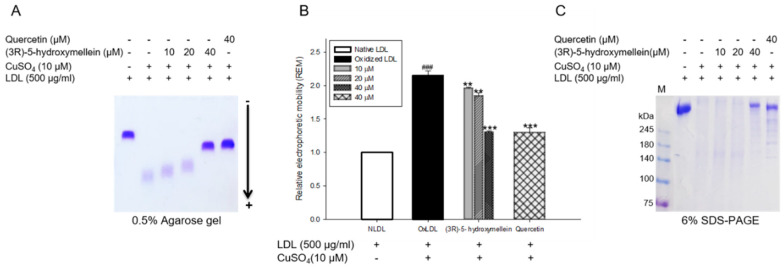
Effect of (3*R*)-5-hydroxymellein on conformational changes in oxidized LDL. (**A**) Electrophoretic mobility profiles of LDL, which were treated with (3*R*)-5-hydroxymellein (at 10, 20, and 40 μM) in the presence of 10 μM of copper ion. (**B**) The graph shows the calculation of relative electrophoretic mobility, which was calculated as the distance traveled from the origin. (**C**) SDS-PAGE of oxidized LDL by 10 μM copper ion for 4 hr in the absence or presence of extract (at 10, 20, and 40 μM). Quercetin used as a positive control. These data are expressed as mean ± SD of three independent experiments. ^###^
*p* < 0.001 vs. NLDL (native LDL); ** *p* < 0.01 vs. oxLDL (oxidized LDL); *** *p* < 0.001 vs. oxLDL (oxidized LDL).

**Figure 4 biomolecules-10-00715-f004:**
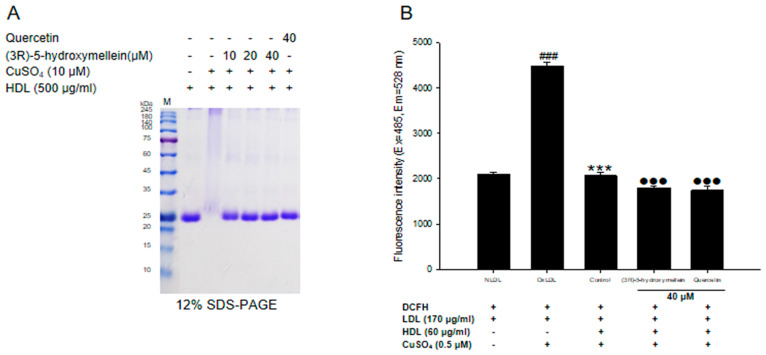
Effects of (3*R*)-5-hydroxymellein on apoA-I aggregation and anti-LDL oxidation in oxidized HDL. (**A**) SDS-PAGE of oxidized HDL by 10 μM copper ion for 4 hr in the absence or presence of extract (at 10, 20, and 40 μM). (**B**) Inhibitory effect of (3*R*)-5-hydroxymellein treated-HDL on LDL oxidation. Quercetin used as a positive control. ^###^
*p* < 0.001 vs. NLDL (native LDL); *** *p* < 0.001 vs. oxLDL (oxidized LDL); ^●●●^
*p* < 0.001 vs. control.

**Figure 5 biomolecules-10-00715-f005:**
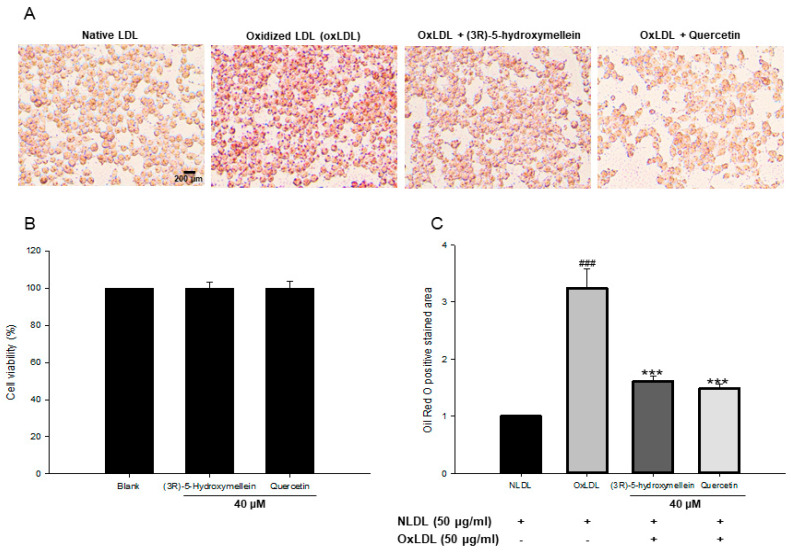
Effects of (3*R*)-5-hydroxymellein on oxLDL induced the lipid accumulation in RAW264.7 cells. (**A**) RAW264.7 murine macrophage cells treated with native LDL, oxLDL, and oxLDL with (3*R*)-5-hydroxymellein were stained with Oil red O solution. Representative images of Oil red O-stained lipid droplets. (**B**) Cytotoxicity test of (3*R*)-5-hydroxymellein in RAW264.7 cells was performed with ELUS cell viability assay kit. (**C**) The Oil Red O positive cells stained area was calculated using the Image J software. Quercetin was used as a positive control. ^###^
*p* < 0.001 vs. NLDL (native LDL); ^***^
*p* < 0.001 vs. oxLDL (oxidized LDL).

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
