# Peer review of "Anti-Atherosclerotic Activity of (3R)-5-Hydroxymellein from an Endophytic Fungus Neofusicoccum parvum JS-0968 Derived from Vitex rotundifolia through the Inhibition of Lipoproteins Oxidation and Foam Cell Formation"

_biomolecules, 2020, doi:10.3390/biom10050715_

Round 1

Reviewer 1 Report

This manuscript is on anti-LDL oxidation activities of 5-hydroxymellein isolated from cultures of an endophytic fungus, Neofusicoccum parvum. Since its inhibitory activity against oxidation of human LDL and HDL was investigated in this research for the first time regardless of many biological activities of 5-hydroxymellein reported to date, it is worth to be published in the journal Biomolecules. However, the followings should be revised or commented by the authors.

Major comments

  • It is wondering if the isolated metabolite (3R)-5-hydroxymellein is cytotoxic. The authors need to address its cytotoxicity.
  • In the experimental section, the identification of the fungus was not indicated. The authors should provide how this fungus was identified.

Minor comments

  • In the section of results, the scale bar should be indicated in figure 5A.
  • Some grammatical issues,Page 290   The multimeric pattern of apoA-I showed in oxidized HDL, which showed that copper ions accelerate the aggregation of apoA-I.
  • Page 237   were -> was
  • The multimeric pattern of apoA-I shown in oxidized HDL indicated that copper ions accelerated the aggregation of apoA-I.
  • Page 321   researched -> researches

Author Response

Thank you very much for your careful review on our manuscript.

According to reviewers’ comments, all comments were answered one by one as shown below. Thus, several parts of the revised manuscript have been corrected and indicated in red color concerning some modification in comparison with the previous manuscript.

This manuscript is on anti-LDL oxidation activities of 5-hydroxymellein isolated from cultures of an endophytic fungus, Neofusicoccum parvum. Since its inhibitory activity against oxidation of human LDL and HDL was investigated in this research for the first time regardless of many biological activities of 5-hydroxymellein reported to date, it is worth to be published in the journal Biomolecules. However, the followings should be revised or commented by the authors.

Major comments

  • It is wondering if the isolated metabolite (3R)-5-hydroxymellein is cytotoxic. The authors need to address its cytotoxicity.

       Answer: Thanks for your good suggestion, we added method and result for        cytotoxicity analysis at lines 206-213, lines 316-317, and legend of figure          5(B), respectively.

  • In the experimental section, the identification of the fungus was not indicated. The authors should provide how this fungus was identified.

        Answer: We added the identification of the fungal strain at lines 110-111.

Minor comments

  • In the section of results, the scale bar should be indicated in figure 5A.

        Answer: According to the comment, we indicated a scale bar in figure 5A.

  • Some grammatical issues,

Page 290 The multimeric pattern of apoA-I showed in oxidized HDL, which showed that copper ions accelerate the aggregation of apoA-I. -> The multimeric pattern of apoA-I shown in oxidized HDL indicated that copper ions accelerated the aggregation of apoA-I.

→ According to the comment, we changed the sentence of “The multimeric pattern of apoA-I showed in oxidized HDL, which showed that copper ions accelerate the aggregation of apoA-I” to “The multimeric pattern of apoA-I shown in oxidized HDL indicated that copper ions accelerated the aggregation of apoA-I”.

Page 237   were -> was

→ According to the comment, “were” was replaced by “was” at lines 246.

Page 321   researched -> researches

→ According to the comment, we changed ‘researched’ to ‘researches’ at line 333.

If you need further information or more clarification, please let me know.

Best regards,

Sang Hee Shim

Professor

College of Pharmacy, Duksung Women’s University

144Gil-33, Samyang-ro, Dobong-gu, Seoul 01369, South Korea

Reviewer 2 Report

Prof. Shim et al. elaborated the alleviating activity of (3R)-5-hydroxymellein on atherosclerosis via inhibition of lipoprotein oxidation as well as foam cell formation. The experimental designs to validate their claims appear to be coherent and logical, leading this reviewer to believe that their study would justify the publication in the Journal. There are a few minor editorial revisions required to perfect the quality of article. Authors are prompted to reaffirm the references (title, style for scientific names, typo, etc) and revise them if necessary-for instance, the title and its style of ref 28.

Author Response

Thank you very much for your careful review on our manuscript.

According to reviewers’ comments, all comments were answered one by one as shown below. Thus, several parts of the revised manuscript have been corrected and indicated in red color concerning some modification in comparison with the previous manuscript.

Prof. Shim et al. elaborated the alleviating activity of (3R)-5-hydroxymellein on atherosclerosis via inhibition of lipoprotein oxidation as well as foam cell formation. The experimental designs to validate their claims appear to be coherent and logical, leading this reviewer to believe that their study would justify the publication in the Journal. There are a few minor editorial revisions required to perfect the quality of article. Authors are prompted to reaffirm the references (title, style for scientific names, typo, etc) and revise them if necessary-for instance, the title and its style of ref 28.

→ According to the comment, we checked the details of all the references and then revised those for references #2, 4, 5, 8, 25, 28, and 48.

If you need further information or more clarification, please let me know.

Best regards,

Sang Hee Shim

Professor

College of Pharmacy, Duksung Women’s University

144Gil-33, Samyang-ro, Dobong-gu, Seoul 01369, South Korea
